# An Investigation of the General Population’s Self-Reported Hand Hygiene Behaviour and Compliance in a Cross-European Setting

**DOI:** 10.3390/ijerph18052402

**Published:** 2021-03-01

**Authors:** Aaron Lawson, Marie Vaganay-Miller, Robert Cameron

**Affiliations:** Belfast School of Architecture and the Built Environment, Ulster University, Newtownabbey BT370QB, UK; m.vaganaymiller@ulster.ac.uk (M.V.-M.); rj.cameron@ulster.ac.uk (R.C.)

**Keywords:** hand hygiene, behaviour, compliance, general population, UK, Spain, travel, survey

## Abstract

Every year, thousands of people from the UK travel to other countries for work and leisure. Europe, and particularly Spain, is one of the most popular travel destinations for people from the UK. However, it is known that travel to other countries can enhance the risk of communicable disease transmission from person to person, especially when a new one emerges. Adequate hand hygiene behaviour and compliance is widely accepted as being a simple, effective method in preventing the spread of communicable diseases that may be contracted during travel abroad. There is a well-established body of work investigating hand hygiene practice and compliance in community settings, but no recent studies have examined the hand hygiene practice and compliance of the general population when travelling abroad or in a cross-European context. The findings of this study indicated that most UK members of the general population when travelling abroad have a good level of understanding of the importance of adequate hand hygiene practice and compliance and its role regarding communicable disease prevention and control. As such, self-reported levels of compliance were high. Similar findings were made for Spanish members of the general population. However, while self-reported perceptions of adequacy of hand hygiene performance were relatively high, particularly among UK respondents, this was not supported by responses specifically focused on hand hygiene behaviour. However, differences in self-reported adequacy regarding the importance of handwashing versus hand drying, the number of steps that should be followed and the length of time that should be spent washing and drying hands were found for each group. This suggests that self-reported compliance may reflect intention to practice hand hygiene rather than true compliance. It also suggests that there are gaps in knowledge regarding the adequate method of hand hygiene among the cohort as a whole, and indeed these differences may account be a factor in for the high transmission rates of communicable disease when travelling abroad.

## 1. Introduction

Current projections suggest that the annual number of international travellers will reach 1.8 billion by 2030 [1]. Europe remained the most common destination for international travellers in 2018, receiving approximately 713 million arrivals, an increase of almost 19% from the previous year [2].

Travelling abroad can increase the risk of communicable disease transmission from person to person due to the large number of people travelling between different countries and physically interacting in close quarters [3,4]. Some studies estimating that between 22% and 64% of international travellers and tourists become ill during or after travel [5,6,7]. Of the limited studies conducted on travel-associated communicable diseases within Europe, it is estimated that between 33% and 35% of them are due to gastrointestinal diseases, a major cause of diarrhoea and the most common reason why tourists seek medical attention when abroad [8,9,10]. Similarly, other epidemiological data revealed that between 2013 and 2018, there were 40,070 reported cases of Shiga toxin-producing Escherichia coli (STEC) and Verocytotoxin-producing Escherichia coli (VTEC) infection across Europe alone, another major cause of diarrhoea and illness within community settings [11]. More specifically, the UK reported 1840 cases in 2018 and Spain reported 126 in the same period [11]. These types of communicable diseases and others are primarily transmitted from person to person via physical touch or via the faecal oral route both directly and indirectly [8,9].

Good hand hygiene behaviour and compliance is recognised as being the most effective method in preventing the spread of such diseases especially when travelling abroad [12,13]. It is important to wash and dry hands well at key times such as after using the toilet, after coughing or sneezing into hands and before preparing or eating food [14,15]. Previous studies suggest that good hand hygiene practice and compliance can reduce the risk of diarrhoeal disease transmission by between 23% and 48% [16,17,18]. This is because the mechanical action of washing hands with water and soap for at least twenty seconds or more, and then drying properly afterwards is shown to effectively remove any communicable pathogens from the surface of the skin [19,20,21,22].

Most research studies examining hand hygiene behaviour and compliance use self-reporting methods like surveys to do so, but the majority of these have been conducted within healthcare settings or within the food industry [22,23]. Of the limited number of survey studies conducted amongst the general population, self-reported compliance is varied, but generally higher compared to observational studies [24,25,26,27,28]. Across Europe, hand hygiene compliance rates are difficult to determine due to the limited data available on the general population. However, a previous survey conducted in 2015 of the general population’s hand hygiene habits after using the toilet estimated that the European hand hygiene compliance rate was around 70% overall, with Bosnians reporting to have the highest compliance rate at 96% and the Dutch the least likely at 50% [29].

Currently, the delivery of most hand hygiene education across Europe is founded in healthcare guidance and is primarily aimed towards healthcare professionals like doctors and nurses, and young children under the age of five years old in nursery school settings [30,31,32,33,34]. A key goal of hand hygiene education is in ensuring that the correct number of hand hygiene steps should be followed at key times and that the correct length of time (at least twenty seconds or more) should be adhered too when washing and drying hands [21,22].

The reasons for differences in the compliance rates of the limited data that is available is difficult to determine. Most hand hygiene behaviour is thought to be inherent in nature and is typically defined at an early age through a mixture of early childhood education on personal hygiene and through the influence of key role models like parents and teachers in reinforcing the behaviour [31,32,33,34]. Therefore, if hand hygiene education at an early age is inadequate, or influence from key role models is inaccurate, then poor behaviour will persist unless otherwise corrected [33,34,35].

Since good hand hygiene compliance is an important behaviour, especially when travelling abroad, understanding the mechanisms underlying this behaviour is important for preventing communicable disease transmission within the general population [34].

As such, no recent studies have examined the hand hygiene behaviours and compliance of the general population when in a cross-European context. This, coupled with the fact that there is little recent epidemiological data available on communicable diseases primarily transmitted between travellers within Europe, makes it difficult to determine the impact of hand hygiene behaviour on communicable disease transmission when travelling abroad. Therefore, the aim of this research was to investigate the hand hygiene behaviour and compliance of the general population in a cross-European setting.

## 2. Materials and Methods

To fulfil the aim of the study; an anonymous, self-reporting, cross-sectional questionnaire was used to survey both UK and Spanish members of the general population in a popular Spanish holiday location on their hand hygiene knowledge, attitudes, practice and compliance when using public restrooms. The design, content and layout of the questionnaire was informed by similar, previously conducted studies carried out amongst the general population in recent years [28,29]. Data was collected concurrently for 10 days in August 2017 by a single researcher. Questions were translated in both languages accordingly. Ethical approval for the study was granted by Ulster University’s Built Environment Research Institute’s (BERI) Research Ethics Filter Committee (Auth. No: 0517).

### 2.1. Selection and Description of Research Subject

Spain was selected as the research location because it is one of the leading holiday destinations in Europe, especially amongst English speakers. It was reported that there were 18.52 million visits by UK tourists to Spain in 2018, the highest number of English-speaking tourists to any European country [36]. A well-known, popular tourist city (Palma de Mallorca) was selected in Spain as the research location. Convenience sampling was used when selecting and recruiting research subjects for the study due to time and resource restrictions. The lead researcher approached members of the general population in Palma de Mallorca’s city centre at the Plaza de España, which is a popular tourist location. This included both English and Spanish-speaking individuals and the purpose of the study was explained in brief to each person. Informed consent was obtained from willing research subjects prior to participation. Each participant was given an Apple iPad tablet device and asked to complete the questionnaire privately in their own time. The exclusion criteria for this study was if a research subject was from a vulnerable group (under 18 years old, those with learning disability or the very elderly), or if they expressed their wish not to participate or cancel their participation at any time. All research subjects were informed that their participation in the questionnaire could be terminated at any time, and no personal details such as names, addresses or medical history would be recorded as per ethical considerations.

### 2.2. Data Analysis

The data collected was analysed using IBM’s SPSS Statistical Software (v.24) (IBM, New York, NY, USA). Descriptive and inferential statistics were performed. Chi-square analysis was used to identify statistically significant comparisons in the hand hygiene behaviours of the general population when using public restrooms. The statistically significant level was accepted as *p* = ≤ 0.05 with Confidence Levels of 95% (CI) reported where applicable.

## 3. Results

In total, 284 members of the general population were surveyed over a 10-day period and there were 148 valid research subjects, a response rate of 52.11%. This included 56 research subjects from the UK and 92 research subjects from Spain.

### 3.1. Self-Reported Hand Hygiene Knowledge

The self-reported hand hygiene knowledge of both UK and Spanish research subjects are presented in Table 1 below. Regarding hand hygiene education, most research subjects 70.27% (UK: 57.14%, ESP: 78.26%) reported that it was a parent/guardian who taught them how to wash their hands. However more Spanish research subjects self-reported being taught how to wash their hands by a Parent/Guardian (78.26%) compared with UK research subjects (57.14%). This was statistically significant (*n* = 148, χ^2^ = 15.66, *p* = ≤ 0.01). Similarly, 17.57% of research subjects overall self-reported that they taught themselves how to wash their hands, with more UK research subjects (32.14%) than Spanish research subjects (8.70%) doing so.

Overall 92.57% (UK: 89.29%, ESP: 94.57%) of research subjects self-reported that handwashing was ‘very important’ after using the toilet while 3.38% (UK: 5.36%, ESP: 2.17%) reported that it was ‘not important at all’. This information is shown in Table 2 below.

Just 62.16% of research subjects (UK: 50.00%, ESP: 69.57%) reported that hand drying was ‘very important’ while 13.51% (UK: 12.50%, ESP: 14.13%) said it was ‘not important at all’ (Table 3). Although, significantly more Spanish research subjects ranked hand drying as being ‘very important’ compared with UK research subjects who were ‘unsure’. This was statistically significant (*n* = 148, χ^2^ = 8.64, *p* = ≤ 0.01).

When research subjects were asked to rate their own hand hygiene compliance compared to others (Table 4), 76.35% (UK: 87.5%, ESP: 69.56%) reported that it was either ‘excellent’ or ‘very good’. No research subjects reported that their hand hygiene compliance compared to others was ‘poor’. Spanish research subjects were statistically less confident rating their compliance compared with UK research subjects (*n* = 148, χ^2^ = 18.50, *p* = ≤ 0.01).

When asked to rank the importance of various infection prevention behaviours(Table 5), with 1 being most important and 7 being least important; UK and Spanish research subjects overall scored ‘adequate hand hygiene’ as first choice at 41.04%, over the other behaviours, and this was statistically significant for both UK and Spanish research subjects (*n* = 148, χ^2^ = 27.59, *p* = ≤ 0.01).

Research subjects were asked about the effectiveness of adequate hand hygiene practice in relation to a number of hygiene-related outcomes (Table 6), with 1 being ‘extremely effective’ and 5 being ‘not effective at all’. Overall, UK research subjects scored ‘Removing visible dirt from my hands’ (Mean: 1.66), ‘Preventing the spread of diseases and germs to others’ (Mean: 1.73), and ‘Making my hands smell nice’ (Mean: 1.82) as being the most effective hand hygiene outcomes. However, Spanish research subjects scored ‘Protecting the health of vulnerable people such as children and the elderly’ (Mean: 1.53), ‘Teaching others good personal hygiene behaviours’ (Mean: 1.60) and ‘Preventing the spread of diseases and germs to others’ (Mean: 1.69) as being the most effective hygiene outcomes. This was statistically significant (*p* = ≤ 0.01).

When asked about the main reason they washed their hands after using the toilet (Table 7), most research subjects selected ‘preventing the spread of diseases and germs to others’ as the being main reason (Total: 87.16%, UK: 83.93%, ESP: 89.13%). ‘To remove visible dirt from my hands’ followed this (Total: 8.78%, UK: 12.50%, ESP: 6.52%), and then ‘protecting the health of vulnerable people such as children and the elderly’ (Total: 2.03%, UK: 1.79%, ESP: 2.17%).

### 3.2. Self-Reported Hand Hygiene Behaviour

The self-reported hand hygiene behaviour of UK and Spanish members of the general population is presented in the next section. Overall 63.45% (UK: 60.71%, ESP: 65.19%) of research subjects answered that they ‘always’ washed their hands after using the toilet. No research subjects reported ‘never’ washing their hands after using the toilet (Table 8).

For hand drying (Table 9), around 63.45% (UK: 60.71%, ESP: 65.19%) of research subjects answered that they ‘always’ dried their hands after using the toilet. About 2.07% (UK: 3.57%, ESP: 1.12%) answered that they ‘never’ dried their hands after using the toilet.

For behaviour related to hand hygiene technique (Table 10), the scoring method utilised determined the number of research subjects who successfully performed all eight recommended hand hygiene steps as outlined in healthcare guidance. This included those who performed the basic number of steps (apply water, soap and dry), those who performed a mixture of steps (basic steps plus various rubbing methods), and those who did not perform any steps after using the toilet (did not wash or dry). Overall, 18.24% (UK: 8.93%, ESP: 23.91%) of research subjects reported performing all eight steps after using the toilet, with 14.19% of research subjects (UK: 1.79%, ESP: 21.74%) not performing the minimum number of steps which included wetting hands with water, applying soap, and drying properly afterwards.

Research subjects were asked to estimate the amount of time they spent washing and drying their hands after using the toilet. For handwashing (Figure 1), the mean length of time self-reported by UK research subjects was 34.82 s and the range of handwashing times reported were between 5 and 51 s. The mean length of time self-reported by Spanish research subjects washing hands was 34.47 s and the range was between 5 and 100 s. There was a statistically significant relationship between the time spent washing hands for both UK and Spanish research subjects in this study (*n =* 148, χ^2^ = 67.81, *p* = 0.02).

In relation to hand drying (Figure 2), the mean length of time self-reported by UK research subjects was 25.16 s (SD: 12.74) and the range was between 6 and 52 s. The mean length of time self-reported by Spanish research subjects drying hands was 24.90 s (SD: 17.91) and the range was between 0 and 100 s. There was a statistically significant relationship between the time spent drying hands for both UK and Spanish research subjects in this study (*n =* 148, χ^2^ = 80.32, *p* = < 0.01).

## 4. Discussion

Good hand hygiene practice and compliance is important for preventing the transmission of communicable diseases especially when travelling abroad. The findings of this study indicated that while the general population have a good understanding of the importance of hand hygiene, they demonstrate a poor level of self-reported hand hygiene knowledge and compliance. While there were some differences in behaviours between UK and Spanish participants, which are discussed later, this general poor knowledge and performance was common across both groups.

Regarding general infection prevention and control knowledge, most UK and Spanish members of the general population were shown to have a good level of knowledge of the importance of good hand hygiene in communicable disease prevention. A total of 92.57% of the respondents recognised that hand hygiene was ‘very important’ after using the toilet and rated their own hand hygiene compliance as ‘excellent’ or ‘very good’ compared to others. In addition, the main self-reported motivations for practicing hand hygiene after using the toilet mostly centred around the protection of self and others, as well as removing visible dirt from hands. Both groups also rated hand hygiene as the most important practice for preventing the spread of disease. However, this self-reported knowledge conflicts significantly when considering reported behaviours. Despite this apparent understanding of the important role good hand hygiene practice can play in preventing the spread of infections and the altruistic motivations for practising good hand hygiene, only 63.45% of the total population then went on to report always washing their hands after going to the toilet. Furthermore, only 35.86% of the population reported drying hands properly. Additionally, 62.16% overall rated hand drying as important, while 24.32% were unsure and 13.51% rated it as not important at all. Adherence to both behaviours are important for effectively removing communicable pathogens from the surface of hands [18,19,20,21]. A mere 18.24% of the sample population indicated that they perform all of the 8 steps required for effective hand hygiene. These reported behaviours also contrast significantly when taking into account that 76.35% of the sample population reported that they rated their hand hygiene compliance in comparison to others as excellent or very good. Higher than the percentage reporting that they always wash their hands after using the toilet.

The conflict between apparent knowledge regarding the public health significance of good hand hygiene practice and the self-reported poor hand hygiene compliance in this study is most likely due to a complex interaction of factors. Firstly, the general population’s self-reported knowledge and positive attitude and motivations for practicing good hand hygiene balanced against a disparity of knowledge on what constitutes good hand hygiene. A lack of knowledge on the adequate method of hand hygiene regarding the importance of practicing all eight recommended hand hygiene steps and spending the correct length of time washing and drying hands, rather than a lack of knowledge on its importance for disease prevention and control. This indicates that most of the general population are aware of the importance of good hand hygiene behaviour and compliance in preventing the spread of communicable diseases, and suggests that they do not just wash their hands when they are visibly soiled, but also to protect vulnerable populations and prevent the spread of disease to others. The motivation is there but the knowledge and understanding of what constitutes good practice is lacking. Previous studies have highlighted the relationship between having a good level of knowledge and good hand hygiene behaviour [31,32,33]. For example, a greater number of respondents viewed handwashing as being ‘very important’ than viewed hand drying as being very important. This is despite good hand drying compliance being essential in good adherence to overall hand hygiene behaviour. Lower expectations and attitudes towards practicing hand drying after washing has been found in previous studies [19,20] and is again indicative of poor knowledge and understanding. Future hand hygiene interventions should focus on improving people’s level of knowledge and understanding of good hand hygiene practice including attitudes on the importance of good hand drying compliance.

Secondly, some elements of self-reported hand hygiene compliance in this study may have been the result of over-reporting by UK and Spanish members of the general population who answered the survey. Over 76% of respondents rated their hand hygiene compliance as very good or excellent as compared to others despite only 63.45% reporting that they always wash their hands after using the toilet. Furthermore, the 18.24% of respondents following all eight hand hygiene steps seems incompatible with the claimed periods of time washing and drying hands in both groups. Responses show that 86% and 91% of Spanish and UK respondents respectively reported taking 20 s or more washing hands. Indeed 35% of Spanish respondents reported taking 35 s or more washing hands. In addition, it is normally difficult for people to self-evaluate personal, inherent behaviour like hand hygiene particularly in terms of timing. Unless there is a prompt available such as an electronic timer then most people naturally overestimate the length of time spent washing and drying hands. Other, observational studies have found much lower mean lengths of time spent washing and drying hands [24,25]. The level of self-reported compliance on aspects such as the quality of hand washing and the time spent washing and drying hands in this study may in fact represent intention to practice good hand hygiene compliance, rather than accurately represent true hand hygiene compliance. The lack of knowledge on appropriate technique again impacting on intent. The over-reporting of socially desirable behaviours like hand hygiene has been reported in previous studies [24,25] and brings into question the use of self-reporting methods like surveys for determining hand hygiene behaviour and compliance.

There were some statistically significant variances within the research. A greater proportion of the Spanish respondents reported having learned hand hygiene from their parents as compared to UK respondents, 78% and 57% respectively. Research indicates that hand hygiene behaviour is defined at an early stage through role models such as parents or guardians which will influence their practice until changed through education. The results of the study may indicate a link between the reported lack of early parental influence, particularly in UK respondents, and the subsequent lack of knowledge and poor practice of good hand hygiene compliance which then persists until corrected [31,32,33,34,35]. Hand hygiene education targeted at parents may have the potential to make significant improvements in hand hygiene performance for present and future generations.

In terms of self-reported knowledge and behaviour, there were some statistically significant findings which again raise questions about the use of self-reporting methods and individual knowledge. A greater number of Spanish respondents reported hand drying as being very important compared to UK respondents, 69.57% and 50% respectively, indicating a self-reported higher level of knowledge of this aspect of good hand hygiene practice among the Spanish sample. Furthermore, a larger number of Spanish respondents reported spending more time both washing and drying hands than UK respondents and significantly 23.91% of Spanish respondents as opposed to 8.93% of UK respondents self-reported using all eight designated steps of good hand hygiene practice. Notwithstanding this 87.5% of UK respondents compared to 69.5% of Spanish respondents rated their hand hygiene compliance as very good or excellent, while conversely fewer UK respondents reported always washing their hands after using the toilet compared to Spanish respondents.

These comparative findings suggest, that while levels of knowledge, understanding and compliance across the cohort are low generally, UK respondents while being more confident in rating their hand hygiene performance compared to others, appear to have knowledge and hand hygiene practices which are significantly lower than the Spanish cohort. This again highlights the knowledge deficit in relation to hand hygiene, and may indicate an increased risk of contracting and spreading infections for UK visitors abroad as compared to Spanish residents.

The overall findings in this study could be indicative of why many communicable diseases, and those common during travel abroad like diarrhoeal disease are easily and readily spread within the general population; because most people’s hand hygiene behaviour is defined at an early age and through the influence of key role models like parents or guardians until it becomes inherent. Therefore, people will continue to wash their hands throughout their life in the manner they believe is correct until they are otherwise told. The differences in self-reported hand hygiene compliance regarding ranking of the importance of hand drying, the number of hand hygiene steps that should be followed and the length of time that should be spent washing and drying hands may allude to the fact that most people are in fact not washing their hands well enough. This theory has been suggested in similar, previous studies [32,33,34].

Some limitations were identified during this study. Due to ethical, time and resource constraints, only one European location could be surveyed and only a single researcher was available to survey members of the general population so there may have been subjective bias in the selection of research subjects as a result. Additionally, the sampling method used may have led to recruitment and selection bias but was chosen due to the strict timeframe for the study completion.

## 5. Conclusions

This study aimed to investigate the self-reported hand hygiene practice and compliance of the general population in a cross-European setting, mainly focusing on UK citizens when abroad in Spain as well as local Spanish people. Both the self-reported hand hygiene behaviour and compliance of UK and Spanish members of the general population were recorded.

The findings indicated that most UK and Spanish members of the general population when abroad have a good level of self-reported knowledge of the importance of adequate hand hygiene practice and compliance in communicable disease prevention and protection of others from disease and illness. However, a gap in knowledge of the adequate method of hand hygiene regarding the key times at which hand hygiene should be practiced, how long hands should be washed and dried for and the number of steps that should be followed was poor. Further differences in attitudes towards the necessity of handwashing versus hand drying were found between both groups.

The findings indicate that current public knowledge of the adequate method of hand hygiene compliance is varied and poor. The findings also suggest that the use of self-reporting methods for determining levels of hand hygiene compliance within a community setting is not very reliable or accurate. These factors, coupled with the large number of travellers between the two countries and across Europe each year, may allude to why and how many communicable diseases are easily and readily transmitted amongst the general population, and across geographical barriers especially when new disease outbreaks occur.

Future research should therefore focus on exploring the reasons for the differences in self-reported hand hygiene behaviour and compliance within the general population in multiple European settings, and whether these accurately reflect true hand hygiene behaviour in a public setting. This information will be useful for public health bodies in both the UK, Spain and the wider European community for tackling highly prevalent communicable diseases.

## Figures and Tables

**Figure 1 ijerph-18-02402-f001:**
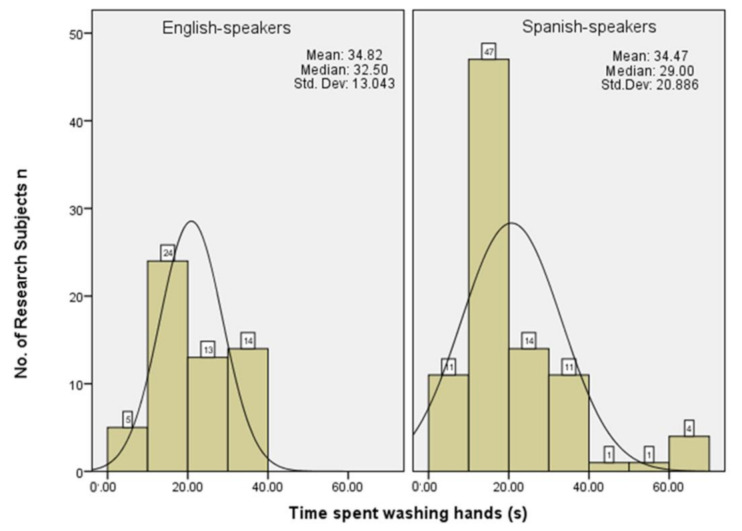
Self-reported handwashing times for UK and Spanish research subjects.

**Figure 2 ijerph-18-02402-f002:**
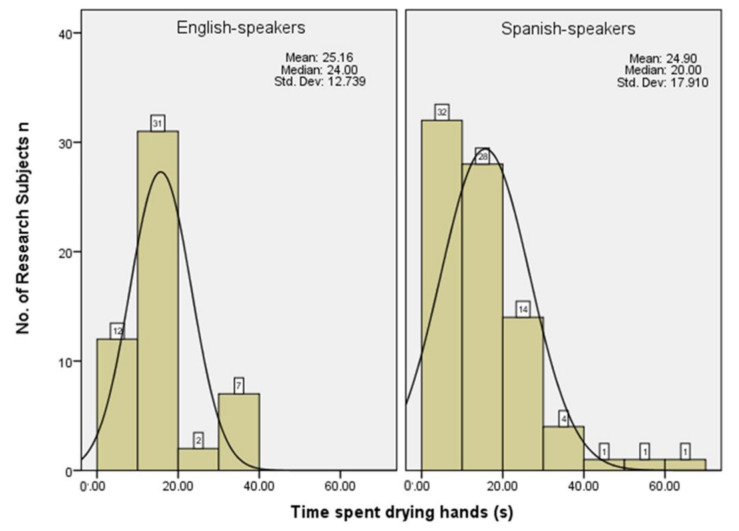
Self-reported hand drying times for UK and Spanish research subjects.

**Table 1 ijerph-18-02402-t001:** Who taught you how to wash your hands properly?

	UK *n* (%)	ESP *n* (%)	Total *n* (%)
Myself	18 (32.14)	8(8.70)	26 (17.57)
Parent/Guardian	32 (57.14)	72 (78.26)	104 (70.27)
Teacher	4 (7.14)	9 (9.78)	13 (8.78)
Friend	0 (0.00)	0 (0.00)	0 (0.00)
Sibling	2 (3.47)	1 (1.09)	3 (2.03)
Other family member	0 (0.00)	2 (2.17)	2 (1.35)
Total	56 (100.00)	92 (100.00)	148 (100.00)

**Table 2 ijerph-18-02402-t002:** In your opinion, how important is handwashing after using the toilet?

	UK *n* (%)	ESP *n* (%)	Total *n* (%)
Very important	50 (89.29)	87 (94.57)	137 (92.57)
Unsure	3 (5.36)	3 (3.26)	6 (4.05)
Not important at all	3 (5.36)	2 (2.17)	5 (3.38)
Total	56 (100.00)	92 (100.00)	148 (100.00)

**Table 3 ijerph-18-02402-t003:** In your opinion, how important is hand drying after using the toilet?

	UK *n* (%)	ESP *n* (%)	Total *n* (%)
Very important	28 (50.00)	64 (69.57)	92 (62.16)
Unsure	21 (37.50)	15 (16.30)	36 (24.32)
Not important at all	7 (12.50)	13 (14.13)	20 (13.51)
Total	56 (100.00)	92 (100.00)	148 (100.00)

**Table 4 ijerph-18-02402-t004:** How do you rate your own hand hygiene compliance compared to others?

	UK *n* (%)	ESP *n* (%)	Total *n* (%)
Excellent	35 (62.50)	25 (27.17)	60 (40.54)
Very good	14 (25.00)	39 (42.39)	53 (35.81)
Good	5 (8.93)	22 (23.91)	27 (18.24)
Fair	2 (3.57)	6 (6.52)	8 (5.41)
Poor	0 (0.00)	0 (0.00)	0 (0.00)
Total	56 (100.00)	92 (100.00)	148 (100.00)

**Table 5 ijerph-18-02402-t005:** Place in order of importance, the following practices in preventing the spread of infection.

Behaviour Ranking	UK%	ESP%	Total%
1. Adequate hand hygiene	66.07	23.08	41.04
2. Vaccination	23.21	24.35	23.88
3. Practicing safe sex	0.00	16.67	9.70
4. Healthy diet & lifestyle	0.00	10.26	5.97
5. Staying at home when ill	7.14	3.85	5.22
6. Use of antibiotics	1.79	6.41	4.48
7. Not sharing personal items	1.79	5.13	3.73
8. Preparing food safely	0.00	5.13	2.99
9. Travel wisely	0.00	5.13	2.99

**Table 6 ijerph-18-02402-t006:** In your opinion, how effective is adequate hand hygiene practice in relation to the following.

	UK Mean Score	ESP Mean Score
Removing visible dirt from my hands	1.66	1.64
Preventing the spread of diseases and germs to others	1.73	1.69
Making my hands smell nice	1.82	2.62
Teaching others good personal hygiene behaviours	2.64	1.60
Protecting the health of vulnerable people	3.25	1.53
To prevent a rise in antibiotic resistance	3.80	2.62

**Table 7 ijerph-18-02402-t007:** What is the main reason you wash your hands after using the toilet?

	UK *n* (%)	ESP *n* (%)	Total *n* (%)
To prevent the spread of diseases and germs to others	47 (83.93)	82 (89.13)	129 (87.16)
To make my hands smell nice	0 (0.00)	1 (1.09)	1 (0.68)
To remove visible dirt from my hands	7 (12.50)	6 (6.52)	13 (8.78)
To prevent a rise in antibiotic resistance	1 (1.79)	0 (0.00)	1 (0.68)
To protect the health of vulnerable people such as children and the elderly	1 (1.79)	2 (2.17)	3 (2.03)
To teach others good personal hygiene behaviours	0 (0.00)	1 (1.09)	1 (0.68)
Total	56 (100.00)	92 (100.00)	148 (100.00)

**Table 8 ijerph-18-02402-t008:** Do you wash your hands after using the toilet?

	UK *n* (%)	ESP *n* (%)	Total *n* (%)
Always	34 (60.71)	58 (65.19)	92 (63.45)
Often	16 (28.57)	26 (29.21)	42 (29.97)
Sometimes	4 (7.14)	4 (4.49)	8 (5.52)
Seldom	2 (3.57)	1 (1.12)	3 (2.07)
Never	0 (0.00)	0 (0.00)	0 (0.00)
Total	56 (100.00)	89 (100.00)	145 (100.00)

**Table 9 ijerph-18-02402-t009:** Do you dry your hands properly after using the toilet?

	UK *n* (%)	ESP *n* (%)	Total *n* (%)
Always	23 (41.07)	29 (32.58)	52 (35.86)
Often	18 (32.14)	35 (39.33)	53 (36.55)
Sometimes	8 (14.29)	15 (16.85)	23 (15.86)
Seldom	5 (8.93)	9 (10.11)	14 (9.66)
Never	2 (3.57)	1 (1.12)	3 (2.07)
Total	56 (100.00)	89 (100.00)	145 (100.00)

**Table 10 ijerph-18-02402-t010:** Which of the following hand hygiene steps do you follow when washing your hands normally?

	UK *n* (%)	ESP *n* (%)	Total *n* (%)
Selected less than 3 steps	1 (1.79)	20 (21.74)	21 (14.19)
Selected 3 key basic steps (water, soap, drying)	5 (8.93)	7 (7.61)	12 (8.18)
Selected a mixture of the 8 steps	45 (80.36)	43 (53.26)	88 (59.46)
Selected all 8 steps	5 (8.93)	22 (23.91)	27 (18.24)
Total	56 (100.00)	92 (100.00)	148 (100.00)

## Data Availability

The data presented in this study are available on request from the corresponding author. The data are not publicly available due to requested restrictions from the Ulster University Research Ethics Filter Committee and Govierno de las Islas Baleares, Mallorca, España.

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
