# Peer review of "An Investigation of the General Population’s Self-Reported Hand Hygiene Behaviour and Compliance in a Cross-European Setting"

_ijerph, 2021, doi:10.3390/ijerph18052402_

Round 1

Reviewer 1 Report

the answers to reviewers' comment are complete and that the manuscript has been implemented satisfactorily.

My advice is to accept the article in this form for publication.

Reviewer 2 Report

I propose to accept MS in its present form

This manuscript is a resubmission of an earlier submission. The following is a list of the peer review reports and author responses from that submission.

Round 1

Reviewer 1 Report

Comments

“An investigation of the general population’s self-reported hand hygiene behavior and compliance in a cross-European setting”

Overall:

The article addresses the important topic with well-constructed methodology. However, the number of samples and sampling method limits this study to make the generalization of hand hygiene behavior based on the findings.

Abstract:

Consider correcting the typos in the abstract.

Introduction:    

The logical flow of the subject matter is good in introduction. Objective is clearly defined.

Methods:

Clear explains the scope and procedure of the study. Consider adding the missing conjunction in the sentence in the section of “selection and description of research subject”

Results: 

There are different things in this section that authors should address.

  1. The second and third sentence in the sub-section “self-reported hand hygiene knowledge and attitudes” gives same repeated information. Consider revising into one.
  2. There are mistake in table numbers embedded within text in many places.
  3. The mismatch of information in the table 6 and the explanation in text for findings of Spanish research subjects.
  4. The description of table 9 is mistake in the text for hand drying component.
  5. Figure 1 is missing in the pdf format

Discussion

Needs revision

Good hand hygiene practice comprises of following major things. I) Handwashing key timings, 2) Steps followed in hand hygiene, 3) Duration of hand hygiene. The findings for above 2 points (refer to explanation in example below) are not at the level to be claimed as good level of compliance.  Hence, the claim made by authors is not correct. Both the aspects and explanation should be added.  

Example 1: Handwashing after toilet is among one of the important critical timings. However, the compliance for this indicator among study samples is only 65%. Hence the claim that general population have good level of self-reported hand hygiene knowledge and compliance may not be correct.

Example 2: The steps in handwashing are important when we study about compliance of hand hygiene behavior. In the present study the practice of handwashing following all the steps (around 8) and selecting among 8 steps is also not at high level. Which in total also accounts only 77%.

Conclusion

It should be revised after the changes in the discussion.

Reviewer 2 Report

The article An investigation of the general population’s self-reported hand hygiene behaviour and compliance in a cross-European setting deals with a very important matter of the hand hygiene. However, the manuscript unfortunately has got a few limitations, some of which are of significant importance.

Specific points:

There is no information concerning the context of dirty hands diseases in Europe, especially in Spain. In the "Introduction" section it is necessary to indicate the epidemiology of these diseases in Spain - the same like in other EU countries or not - are incidence rates among tourists known?

The manuscript lacks the "figure 1".

The "figure 2" is not informative, among others description and statistical significance information should be added.

There is no information about the bioethical committee: who allowed to use the data in the survey, what is the number of the authorization

Reviewer 3 Report

In manuscript IJERPH-820579 authors presented the results of a study investigating hand hygiene practice and compliance of the general population, though a survey performed on 284 subjects recruited in a popular tourist location. Results outlined that most of the subject had a good level of hand hygiene knowledge and attitude towards practicing it, even if knowledge of some specific issues (e.g., number of steps and length of time for a proper hand hygiene practice) showed differences in sub-groups of the enrolled population.

Overall, the issue of potential contagion by contact is topical in these days and the manuscript is written in a clear exhaustive manner and with scientific soundness. However, the objectives, and the obtained results are not very interesting and do not represent an advancement of the current state of knowledge. The introduction provides sufficient elements to characterize the background and the problem statement. The study design is quite poor since the study refers to an extremely specific population of UK citizens on holiday or Spanish citizens residing in the locality. Although the comparison between these two components is interesting, the results seem difficult to transfer to other realities. The results obtained were presented in a clear and complete manner, anyway. Overall, however, I don't think the manuscript has the potential to be published on IJERPH